# Controlled human infection with *Neisseria lactamica* in late pregnancy to measure horizontal transmission and microbiome changes in mother–neonate pairs: a single-arm interventional pilot study protocol

Anastasia A Theodosiou [1], Jay R Laver,[1] Adam P Dale [1], David W Cleary,[1] Christine E Jones [2], Robert C Read [1]

¹Clinical and Experimental Sciences, University of Southampton, Southampton, UK
²Faculty of Medicine and Institute for Life Sciences, University of Southampton, Southampton, UK

**Correspondence to**
Dr Anastasia A Theodosiou; at1u17@soton.ac.uk

## ABSTRACT

**Introduction** Infant upper respiratory microbiota are derived partly from the maternal respiratory tract, and certain microbiota are associated with altered risk of infections and respiratory disease. *Neisseria lactamica* is a common pharyngeal commensal in young children and is associated with reduced carriage and invasive disease by *Neisseria meningitidis*. Nasal inoculation with *N. lactamica* safely and reproducibly reduces *N. meningitidis* colonisation in healthy adults. We propose nasal inoculation of pregnant women with *N. lactamica*, to establish if neonatal pharyngeal colonisation occurs after birth, and to characterise microbiome evolution in mother–infant pairs over 1 month post partum.

**Methods and analysis** 20 healthy pregnant women will receive nasal inoculation with *N. lactamica* (wild type strain Y92-1009) at 36–38 weeks gestation. Upper respiratory samples, as well as optional breastmilk, umbilical cord blood and infant venous blood samples, will be collected from mother–infant pairs over 1 month post partum. We will assess safety, *N. lactamica* colonisation (by targeted PCR) and longitudinal microevolution (by whole genome sequencing), and microbiome evolution (by 16S rRNA gene sequencing).

**Ethics and dissemination** This study has been approved by the London Central Research Ethics Committee (21/PR/0373). Findings will be published in peer-reviewed open-access journals as soon as possible.

**Trial registration number** NCT04784845.

## STRENGTHS AND LIMITATIONS OF THIS STUDY

⇒ The Neisseria lactamica controlled human infection model is well-characterised, with safety, dosing, immunological and genomic data from over 400 adult participants to date.
⇒ This is the first prospective study of horizontal respiratory commensal acquisition in neonates following controlled human infection in pregnancy.
⇒ Longitudinal sampling of multiple respiratory niches in mother–infant pairs allows for microbiome characterisation over time.
⇒ The small sample size may be insufficient to demonstrate a significant association between N. lactamica inoculation and microbiome changes.
⇒ No causal association between N. lactamica inoculation and observed microbiome changes can be concluded in the absence of a control group.

globally, 492 000 and 89 000 infants die each year due to respiratory tract infection (RTI) and meningitis, respectively.[1] While many infant infections are vaccine-preventable, global access to and acceptance of vaccines remains incomplete, such that 1.5 million vaccine-preventable childhood deaths occur each year.[2] Furthermore, existing vaccines do not protect against all important childhood respiratory pathogens (such as respiratory syncytial virus and some serotypes of *S. pneumoniae*), and vaccinations are not routinely given in the first 2 months post partum, when risk of invasive infection is highest. Rising global antimicrobial resistance compounds these challenges, with 214 000 neonates dying of resistant infections each year.[3]

Recent research has highlighted the relationship between URT pathobionts, childhood disease and the evolving infant

## INTRODUCTION

### Upper respiratory pathobionts and infant disease

Upper respiratory tract (URT) pathobionts are common colonising bacteria capable of causing disease in immunocompetent individuals (eg, *Neisseria meningitidis, Streptococcus pneumoniae, Haemophilus influenzae*). Although harmless in many hosts, these remain a leading cause of global childhood death and disease, especially in the first year of life:

microbiome (the site-specific total microbial community).[4] Taxonomic profiling (eg, by bacterial 16S rRNA gene sequencing) demonstrates distinct URT microbiota during acute RTI, and that URT microbiome perturbation is associated with RTI severity and recurrence, childhood chronic wheeze and early allergic sensitisation.[5–8]

Infant RTIs are often preceded by micro changes, including increased relative abundance of URT pathobionts and loss of topography (blurring of distinction between microbiomes at adjacent anatomical sites),[9] suggesting a possible causal role in developing infection. However, as up to 93% of infants are colonised with at least one URT pathobiont at any time,[10] pathobiont colonisation alone is not necessarily associated with infection: rather, the presence of pathobionts may indicate a state of dysbiosis (microbial dysregulation), in which disease-causing bacteria and viruses can cause infection in a previously resilient ecological niche.

Longitudinal cohort microbiome studies suggest that infant URT flora are acquired at least in part from their mothers: there is greater overlap between a neonate's oral flora and that of its own mother than unrelated mothers, and shared bacterial strains (detected by whole genome sequencing) are more long-lived in an infant's mouth than non-maternal strains.[11] Moreover, a subset of maternal oral strains account for a disproportionately large share of their infants' oral flora.[11]

## Can upper respiratory microbiota be manipulated to prevent infant disease?

In the future, it is possible that simple methods of modifying the infant nasopharyngeal microbiome could provide useful adjunct protection in the early months of life.[12] Previously, neonatal nasal inoculation with low-virulence *Staphylococcus aureus* strain 502A has been shown to reduce colonisation by high-virulence *S. aureus* strain 80/81 in response to an outbreak,[13] while pharyngeal inoculation with strain 215 alpha-haemolytic streptococci reduced pharyngeal pathobionts (*Escherichia coli, Klebsiella pneumoniae, S. aureus*) in neonatal intensive care inpatients.[14] Furthermore, a variety of probiotics (living microorganisms administered to confer a health benefit) have been trialled in children and adults, with some reporting a reduction in upper RTI incidence and duration.[15] However, high quality efficacy and microbiome data is lacking, with significant inter-study variation in the bacterial strain (usually lactobacilli, streptococci and bifidobacteria), preparation (usually capsules or dairy supplement), dose and regimen. That being said, there is evidence that many probiotics are safe in pregnancy, neonates and lactating mothers, with some already available over-the-counter to these groups.[16]

## Controlled human infection with *Neisseria lactamica*

Unlike probiotic studies (which often employ proprietary food supplements with uncontrolled concentrations of bacteria), controlled human infection offers a robust model for studying the impact of a defined bacterial inoculum on a human host. Our research group has previously developed a safe and reliable controlled human infection model using nasal inoculation with $10^4$–$10^5$ colony-forming units (CFU) *N. lactamica* strain Y92-1009 in healthy adults.[17 18] In randomised blinded placebo-controlled studies, we have characterised *N. lactamica* colonisation density, duration, cellular and humoral immune responses, and genomic microevolution,[19] observing colonisation rates of up to 85% and no serious adverse reactions following over 400 inoculations.

*N. lactamica* is a common non-capsulated pharyngeal commensal that does not cause invasive disease in immunocompetent hosts. Colonisation peaks at over 40% in 1–2 year-old children,[20] before falling to less than 10% in adulthood,[21] and *N. lactamica* carriage correlates inversely with *N. meningitidis* colonisation.[22] Indeed, *N. lactamica* controlled human infection in healthy adults reduced naturally-acquired *N. meningitidis* carriage from 18% to 8% (both by displacing existing colonisation and preventing new acquisition), exceeding the typical impact of glycoconjugate ACWY vaccination.[17] This displacement effect is not specific to a particular serogroup, unlike available glycoconjugate meningococcal vaccines. Furthermore, natural colonisation with *N. lactamica* has been observed to correlate inversely with invasive meningococcal disease.[21]

## Rationale for this study

The aim of this study is to establish if nasal inoculation with *N. lactamica* in pregnancy results in neonatal colonisation after birth, and to characterise the impact on the developing neonatal URT microbiome. If successful, this will provide a novel model for inducing and studying person-to-person commensal acquisition in neonates (unlike traditional controlled human infection models, which capture inoculation-induced colonisation). Looking ahead, this may provide a new strategy for reducing *N. meningitidis* colonisation and disease in neonates, and of modifying neonatal microbiome development with a view to improving health outcomes.

## METHODS AND ANALYSIS
### Study overview

In this prospective controlled human infection study, 20 healthy pregnant women will receive nasal inoculation with $10^5$ CFU *N. lactamica* (wild type strain Y92-1009) at 36+0 to 37+6 weeks gestation (figure 1). Upper respiratory samples and breastmilk will be collected from mother–infant pairs 1 day, 1 week and 1 month post partum, as well as an umbilical cord blood sample at delivery and an infant venous blood sample at 1 month post partum. Adverse events and relevant clinical data (including mode of delivery, infant feeding and use of antimicrobials) will be recorded. Samples will be analysed by microbiological culture, targeted PCR and sequencing to determine *N. lactamica* Y92-1009 presence, colonisation density and longitudinal microevolution, as well as 16S rRNA gene

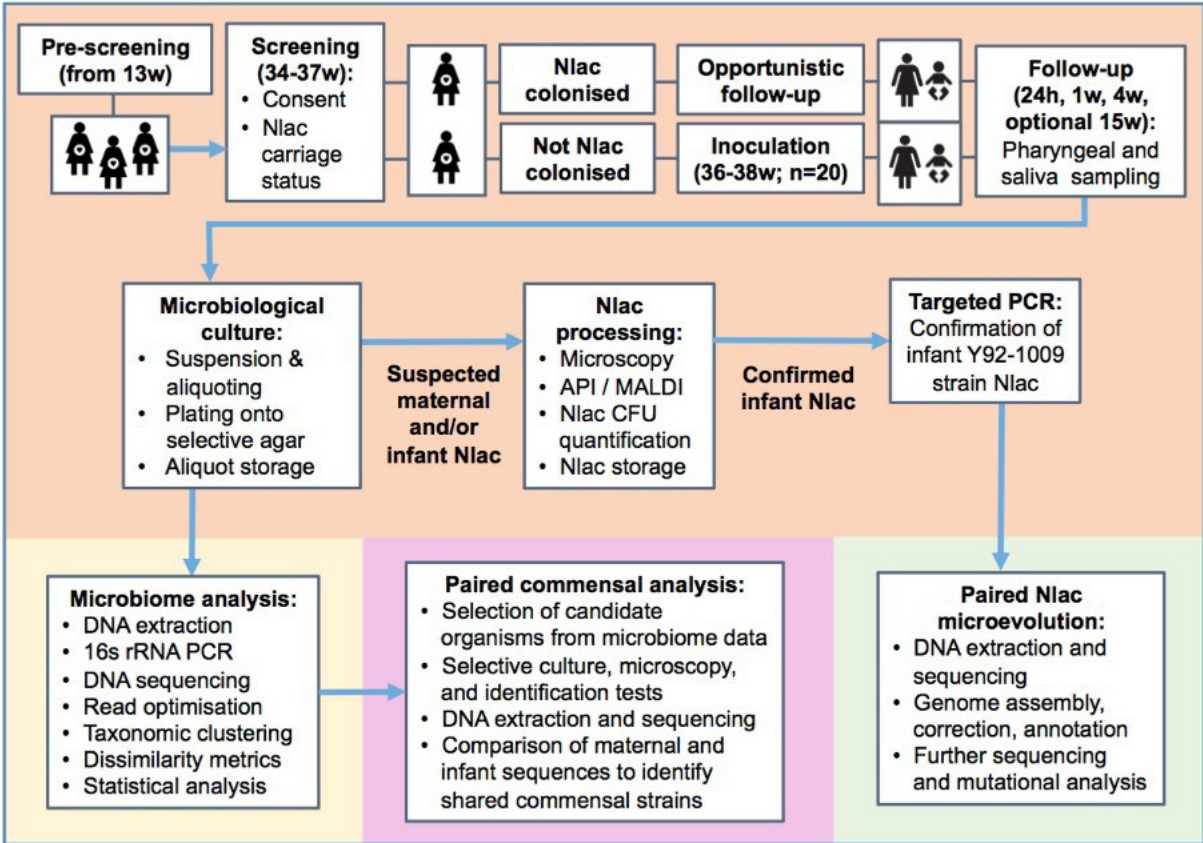

**Figure 1** Study overview flowchart. API, analytical profile index; CFU, colony forming units; h, hours; m, months; MALDI, matrix-assisted laser desorption/ionisation; Nlac, *Neisseria lactamica*; w, weeks.

sequencing to characterise microbiome evolution in mother–neonate pairs.

### Study hypothesis, objective and endpoints

We hypothesise that nasal inoculation of pregnant women with $10^5$ CFU *N. lactamica* strain Y92-1009 will result in maternal upper respiratory *N. lactamica* colonisation; and that horizontal transmission of upper respiratory *N. lactamica* will occur from experimentally-colonised women to their infants by 4 weeks post partum.

This study aims to investigate this hypothesis, and to characterise the safety, colonisation kinetics and microbiome impact of nasal *N. lactamica* inoculation in pregnancy on mother–neonate pairs (table 1).

### *N. lactamica* inoculum

Vials of lyophilised (freeze-dried) *N. lactamica* (LyoNlac) will be produced and stored under Good Manufacturing Practice (GMP)-like standards at the University of Southampton. The strain used for lyophilisation (Y92-1009, sequence type 3493, clonal complex 613) originates from the current GMP-compliant pharmaceutical manufacturing facilities at Public Health England (Porton Down, UK). Lyophilised inoculum stocks can be stored

**Table 1** Primary, secondary and safety study endpoints

| | |
|---|---|
| **Primary endpoint**<br>1. Confirmation of neonatal (0–31 days) *Neisseria lactamica* colonisation by selective culture of nasopharyngeal and saliva samples. | **Secondary endpoints**<br>1. Confirmation of neonatal *N. lactamica* Y92-1009 colonisation by strain-specific PCR.<br>2. *N. lactamica* colonisation density in inoculated volunteers compared with their infants across study time points and sample niche.<br>3. Microbiome composition across study time points and sample niche in inoculated volunteers compared with their infants, and in colonised compared with uncolonised infants.<br>4. *N. lactamica* genome sequence for isolates derived from inoculated volunteers compared with their infants across study time points and sample niche. |
| **Safety endpoints**<br>1. Percentage of inoculated volunteers with adverse reactions or serious adverse events within the study period.<br>2. Percentage of neonates with serious adverse events within the study period. | |

at 4°C (unlike frozen stocks requiring storage at −80°C). LyoNlac will be reconstituted in phosphate buffered saline and diluted to a concentration of $10^5$ CFU/mL for inoculation. In our previous work, this dose resulted in *N. lactamica* colonisation in 100% of 10 adults receiving lyophilised inoculum and 85% of 20 adults using frozen inoculum (ie, 90% overall). The inoculum will be administered immediately following reconstitution and serial dilution, and after no more than 30 min delay; recovery of viable bacteria remains over 70% after 30 min incubation in room air (unpublished data).

### Recruitment of study volunteers

All potentially eligible healthy adult volunteers in the second and third trimesters of pregnancy will be identified by research midwives using a clinical database, and will be sent an ethically-approved study advertisement and a Participant Information Sheet. Individuals who express an interest will be invited to attend a screening visit. Participants will be offered reimbursement for their time, travel and inconvenience.

### Study visits and procedures
#### Screening

Interested individuals will be reviewed by the study doctor at a screening visit (34+0 to 36+6 weeks gestation), to discuss the study timeline and procedures (table 2) and assess eligibility (box 1). Eligible individuals who provide informed consent at screening will be considered enrolled. A detailed medical history will be taken, followed by pharyngeal and saliva sampling to assess for *N. lactamica* carriage; enrolled volunteers who are already naturally colonised with *N. lactamica* will not be inoculated. However, they will be invited to remain in the study as an opportunistic comparison cohort, and will complete all postpartum visits and sampling as for the inoculated mother–neonate pairs (table 2).

#### Inoculation

Eligible consenting volunteers who are not already naturally colonised with *N. lactamica* will receive nasal inoculation with $10^5$ CFU *N. lactamica* Y92-1009, using a sterile pipette (0.5 mL inoculum per nostril).

#### Postpartum follow-up

Mother–neonate pairs will be reviewed 1 day, 1 week and 1 month post partum. At each visit, maternal consent for neonatal sampling will be re-affirmed, and respiratory samples will be collected (table 2). An optional umbilical cord blood sample (visit 3) and neonatal venous blood sample (visit 5) will be collected, as well as relevant clinical data (including mode of delivery, infant feeding and use of antimicrobials). Optional breastmilk samples will also be collected at each postpartum visit (up to 5 mL each at visits 3 to 5). Volunteers may decline optional blood and breastmilk sampling without affecting their enrolment in or completion of the rest of the study.

Volunteers' experiences of participation (including acceptability and tolerability of inoculation and maternal and infant sampling) will be captured using optional questionnaires, to help us identify areas for improving volunteer experience in current and future studies.

### Optional visit 6 (15+/−2 weeks post partum)

At the time of writing, a study amendment is in progress to include an optional study visit at 15±2 weeks post partum (visit 6). This visit will involve maternal and infant pharyngeal, saliva and venous blood sampling, as well as saliva and oropharyngeal swabs from household contacts under 5 years old. In a recent *N. lactamica* controlled human infection study involving non-pregnant adults, anti-*N. lactamica* immunoglobulin G (IgG) (peak titre and titre increment) correlated inversely with subsequent pharyngeal colonisation density at 28 days post-inoculation (unpublished data), suggesting that IgG may play a role in controlling *N. lactamica* carriage. Total IgG is up to 1.5 times more concentrated in umbilical cord blood compared with maternal blood,[23] and the half-life of transplacental maternal IgG is approximately 30 days[24]; thus, infant anti-*N. lactamica* IgG is predicted to be less than 10%–20% that of maternal IgG by 15±2 weeks (3 to 4 half-lives). Visit 6 therefore offers a valuable opportunity to investigate the association between anti-*N. lactamica* IgG and *N. lactamica* carriage in mother–infant pairs.

### Sample processing
#### *N. lactamica* identification

Working in a class 2 microbiological safety cabinet in a Containment Level 2 laboratory, all respiratory samples will be suspended in 1 mL storage medium (10% glycerol in 0.1% diethylpyrocarbonate (DEPC)-treated water). Agar selective for *N. lactamica* (GC agar supplemented with 5-bromo-4-chloro-3-indolyl-β-galactopyranoside (X-gal); Thermo Fisher Scientific, Basingstoke, UK) will be inoculated with sample solution, and all remaining sample solution will be split into 200 μL aliquots for storage in a secure, temperature-monitored −80°C freezer. Following 24–48 hours incubation, colonies with typical *N. lactamica* morphology (blue on GC X-gal agar) will undergo identification by oxidase reaction, Gram stain, microscopy and analytical profile index (API-NH, bioMérieux, Lyon, France) or matrix-assisted laser desorption/ionisation time-of-flight (MALDI-TOF; Bruker, Coventry, UK). *N. lactamica* density will be quantified and isolates will be stored at −80°C. Breastmilk will also be plated onto selective agar to examine for evidence of *N. lactamica*, and residual samples will be stored at −80°C for microbiome analysis.

#### *N. lactamica* Y92-1009 strain-specific PCR

We have identified two chromosomal loci present in *N. lactamica* strain Y92-1009 that, when amplified in conventional multiplex PCR and analysed by agarose gel electrophoresis, produce amplicons with a characteristic band pattern.[18] Using the same multiplex PCR on isolates of naturally circulating strains of *N. lactamica*, this band pattern has not been reproduced. A third locus, present

**Table 2** Study visits and activities

| Timeline | Screening (visit 1) 34+0 to 36+6 weeks gestation | Inoculation (visit 2) 36+0 to 37+6 weeks gestation | Birth (visit 3) 0–24 hours post partum | Follow-up (visits 4 and 5) 7±3 days post partum | 28±3 days post partum | Optional (visit 6) 15±2 weeks post partum |
|---|---|---|---|---|---|---|
| Participant Information Sheet and consent form | + | | | | | |
| Clinical review | + | + | + | + | + | + |
| *Neisseria lactamica* nasal inoculation | | + (Unless already colonised) | | | | |
| Nasopharyngeal swabs (mother) | + | + | + | + | + | + |
| Oropharyngeal swabs (mother) | + | + | + | + | + | + |
| Saliva swabs (mother) | + | + | + | + | + | + |
| Nasopharyngeal swabs (infant) | | | + | + | + | + |
| Saliva swabs (infant) | | | + | + | + | + |
| Umbilical cord blood | | | Optional | | | |
| Infant venous blood (2 mL) | | | | | Optional | Optional |
| Maternal breast milk | | | Optional | Optional | Optional | |
| Maternal venous blood (5 mL) | | | | | | Optional |
| Oropharyngeal and saliva swabs (under 5 years) | | | | | | Optional |

---

**BOX 1    INCLUSION AND EXCLUSION CRITERIA**

**INCLUSION CRITERIA (ALL MUST APPLY)**

⇒ Healthy adult aged 18 years or over on the day of enrolment.

⇒ Singleton pregnancy, 34+0 to 36+6 weeks gestation on the day of enrolment.

⇒ Documentation of a 20-week ultrasound scan with no life-limiting congenital anomalies, and no maxillofacial, otorhinolaryngological or neuroanatomical anomalies.

⇒ Able and willing (in the Investigator's opinion) to comply with all study requirements.

⇒ Able and willing to give written informed consent to participate in the study.

⇒ Booked to receive antenatal care at University Hospital Southampton NHS Foundation Trust (including any associated community antenatal care facilities), UK.

**EXCLUSION CRITERIA (NONE MUST APPLY)**

⇒ Any confirmed or suspected immunosuppressive or immunocompromised state, including: HIV infection; asplenia; recurrent severe infections; or use of immunosuppressant medication (for more than 14 days within the past 6 months, excluding topical and inhaled steroids).

⇒ Planned use of immunosuppressant medication in later pregnancy or post partum.

⇒ Occupational, household or intimate contact with any immunosuppressed persons.

⇒ Participation within the last 12 weeks in a clinical trial involving receipt of an investigational product, or planned use of an investigational product during the study period.

⇒ Prior participation at any time in research studies involving inoculation with N. lactamica.

⇒ Use of oral or intravenous antibiotics within 30 days prior to the inoculation visit.

⇒ Planned use of oral or intravenous antibiotics at any time during the study period (eg, for planned elective caesarean section or group B streptococcus colonisation).

⇒ Allergy to soya or yeast.

⇒ Previous stillbirth or neonatal death.

⇒ Pre-pregnancy diabetes mellitus.

⇒ Any other finding that may (in the Investigator's opinion): increase the risk to the volunteer (or their fetus/infant or close contacts) of participating in the study; affect the volunteer's ability to participate in the study and complete follow-up; or impair interpretation of study data.

---

in all known strains of *N. lactamica*, is included to differentiate between PCR failure (no bands produced) and any non-Y92-1009 strain of *N. lactamica* (band produced, but Y92-1009-specific bands pattern not present). This PCR assay has been validated using genomic DNA and boiled bacterial lysate as a template, with an optimum annealing temperature 65°C and extension time 15 s (unpublished data).

## Microbiome analysis

Paired mother–infant frozen sample aliquots will be transported to the University of Edinburgh Centre for Inflammation Research for 16S rRNA gene sequencing.[6]

Resulting sequence reads will undergo taxonomic classification to compare mothers and infants, and (assuming some neonates become colonised with *N. lactamica*) colonised and uncolonised infants, across different sample niches and study time points.

### *N. lactamica* sequencing

If any neonates become colonised with *N. lactamica*, paired mother–infant *N. lactamica* isolates will undergo whole genome sequencing to assess for differences in *N. lactamica* microevolution in mother–infant pairs. Resulting sequences will be mapped to a closed *N. lactamica* Y92-1009 reference genome for comparison.

## Sample size

To our knowledge, this proof-of-concept represents the first respiratory controlled human infection study in pregnancy. As such, it is not possible to accurately define the sample size needed to conclusively rule out induced horizontal transfer of *N. lactamica*. A pragmatic sample size of 20 inoculated volunteers has been selected: based on our earlier work, *N. lactamica* colonisation is expected in 18 (90%) of volunteers inoculated with $10^5$ CFU *N. lactamica*. This sample would therefore fail to detect horizontal transfer only if it occurs in less than 1 in 18 infants (<6%), in which case it would be impractical and even unethical to pursue horizontal *N. lactamica* controlled human infection as a method of studying and manipulating infant commensalisation. By 1 month of age, 2.3% of infants naturally carry *N. lactamica*.[17] Based on a plot of lower confidence limits against observed infant *N. lactamica* colonisation for a sample of 18 experimentally-colonised volunteers, infant *N. lactamica* colonisation greater than 22% (95% CI (2.3 to 41.7%)) would demonstrate that inoculation-induced carriage exceeds natural *N. lactamica* carriage (figure 2).

We expect to screen at least 24 volunteers to reach our target of 20 inoculations, allowing for 10% dropout and 10% baseline natural *N. lactamica* carriage.[18] It is not clear how maternal or neonatal antibiotics may influence *N. lactamica* carriage and microbiota. At the Principal Investigator's discretion, an additional volunteer may be enrolled and inoculated for every volunteer (or

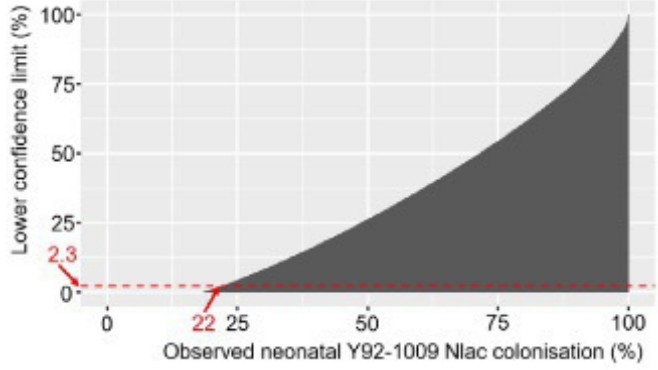

**Figure 2** Lower confidence limit plot for inoculation-induced *Neisseria lactamica* (Nlac) colonisation.

neonate) who receives antibiotics after inoculation, up to a maximum of 10 additional volunteers.

## Statistical analysis

The primary endpoint will be expressed as a proportion N/M, where:

▸ M=Number of mothers colonised with *N. lactamica*.
▸ N=Number of neonates (aged 0–31 days) colonised with the same strain of *N. lactamica* as their own mother.
▸ 'Colonised'=evidence of *N. lactamica* by culture of upper respiratory swabs (using selective media and confirmation by API or MALDI-TOF) on at least one visit.
▸ 'Same strain'=confirmed using Y92-specific *N. lactamica* PCR for inoculated volunteers, or whole genome sequencing for volunteers already naturally carrying *N. lactamica* at screening.

Subgroup analysis will be performed, to compare inoculated mother–infant pairs with naturally colonised mother–infant pairs, and to compare pairs where the mother or infant received antibiotics with pairs not receiving antibiotics.

Specific bioinformatic analysis strategies will depend on the quantity and quality of genomic data produced during the study, and on the rates of *N. lactamica* colonisation observed. Broadly speaking, microbiome bioinformatic pipelines (eg, QIIME2 and R packages) will be used to perform taxonomic classification, calculation of alpha and beta diversity and data visualisation.

## Patient and public involvement

During early study design, an interview-based patient and public involvement project was conducted with 12 pregnant women (unpublished data). All 12 women expressed approval that the proposed study should go ahead, 11 of whom would personally have liked to participate. These discussions led to changes in study design (such as offering follow-up visits in the volunteer's own home, and obtaining an umbilical cord rather than an infant venous blood sample at birth). During participant recruitment following study completion, the University of Southampton Public Engagement Research Unit will assist in showcasing the study at public engagement events.

## ETHICS AND DISSEMINATION
### Ethics and safety

Volunteer safety and ethical study conduct is of paramount importance. This study has been approved by the NHS Research Ethics committee and the Health Research Authority, and is sponsored by the University of Southampton. An External Safety Committee has been appointed to provide independent safety oversight. Safety of *N. lactamica* Y92-1009 inoculum is well-characterised: over 400 adults have previously been inoculated with this *N. lactamica* strain (of whom 30 received lyophilised inoculum, and 54 received $10^5$ CFU), with no serious adverse

reactions reported to date[15] (unpublished data). Due to the use of yeast-based and soya-based products in inoculum production, individuals reporting allergies to yeast or soya will not be eligible for enrolment. Inoculation will take place in a clinical area with access to adult, obstetric and paediatric emergency services. Volunteers will be reviewed by a study doctor immediately prior to and 30 min after inoculation.

Adverse events will be recorded throughout the study, and volunteers will be encouraged to report any concerns (including by telephone to the study doctor out-of-hours). Adverse events will also be recorded for any inoculated withdrawn volunteers, by telephone review at 4 weeks post partum. As per the current Good Clinical Practice guideline,[25] adverse events are any untoward medical occurrence in a volunteer or their neonate, temporally, but not necessarily causally, related to study participation. Delivery of routine peripartum care is not considered an adverse event. Serious adverse events include hospitalisation (or prolongation of hospitalisation for routine peripartum care), persistent or significant disability, a life-threatening occurrence or death from any cause.

Mitigation strategies are in place to minimise the risk of SARS-CoV2 transmission to participants and study staff. Where possible, study visits will be conducted in the volunteer's home, practicing social distancing and appropriate personal protective equipment for non-aerosol generating procedures. If participants report symptoms of or contact with anyone with suspected or confirmed SARS-CoV2 infection, study visits will be rescheduled until testing or self-isolation as necessary.

## Data management and dissemination

All data will be stored in link-anonymised format using paper case report forms stored securely at the research site. Fully anonymised sequencing data will be uploaded to a curated online data repository. Research findings will be published in peer-reviewed journals as soon as is practicable, with an online link to the final approved study protocol. Participants will be provided with a lay summary of the study results once available.

## Potential benefits and study impact

To our knowledge, this is the first respiratory controlled human infection study in pregnancy, and, if successful, would be the first to demonstrate induced person-to-person commensal transmission. This novel model could be used to characterise microbiome and immunological changes associated with induced commensalisation. Inducing *N. lactamica* colonisation in neonates may also provide a new strategy for reducing *N. meningitidis* colonisation and even disease in infants. Although many neonates are protected by trans-placental maternal antibodies, invasive meningococcal disease does rarely occur in infants, especially before vaccination.[26] Looking ahead, improved understanding of microbiome evolution following controlled human infection is necessary before

inoculation in neonates (rather than in pregnant adults) can be considered.

Our team have recently trialled nasal inoculation in healthy non-pregnant adults using genetically-modified (GM) *N. lactamica* that expresses the meningococcal antigen *Neisseria* adhesin A (NadA).[27] This GM inoculum has proved safe and effective at eliciting immune responses to NadA, providing a proof-of-concept that a harmless commensal may be used as a vehicle for prolonged mucosal exposure to an antigen of interest. Looking ahead, GM *N. lactamica* expressing antigens from other clinically-important paediatric respiratory pathogens could be explored, to complement existing childhood vaccination.

**Contributors** AAT prepared the manuscript, led the study design and is the principal investigator and overall guarantor for the study. CEJ and RCR are the chief investigators and provided clinical and academic supervision. APD was involved in clinical and microbiological study design, DWC supervised genomic and bioinformatic study design and JRL supervised laboratory and molecular study design. All authors reviewed, edited and approved the final manuscript.

**Funding** This work is supported by a Medical Research Council Clinical Research Training Fellowship (grant number MR/V002015/1) awarded to AAT. RCR is an NIHR Senior Investigator. The funders have not had input into the design or execution of the study or the preparation of this manuscript.

**Competing interests** None declared.

**Patient and public involvement** Patients and/or the public were involved in the design, or conduct, or reporting, or dissemination plans of this research. Refer to the Methods section for further details.

**Patient consent for publication** Not applicable.

**Provenance and peer review** Not commissioned; externally peer reviewed.

**ORCID iDs**
Anastasia A Theodosiou http://orcid.org/0000-0002-0096-4825
Adam P Dale http://orcid.org/0000-0001-8163-7481
Christine E Jones http://orcid.org/0000-0003-1523-2368
Robert C Read http://orcid.org/0000-0002-4297-6728

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
