## [Reviewer comments · BMJ Open]

ARTICLE DETAILS

TITLE (PROVISIONAL)	Controlled human infection with Neisseria lactamica in late pregnancy to measure horizontal transmission and microbiome changes in mother-neonate pairs: a single-arm interventional pilot study protocol
AUTHORS	Theodosiou, Anastasia; Laver, Jay; Dale, Adam; Cleary, David; Jones, Christine; Read, Robert

VERSION 1 – REVIEW

REVIEWER	Melanie Bissessor University of Melbourne, Department of Public Health
REVIEW RETURNED	07-Oct-2021

GENERAL COMMENTS	Thank you for the interesting manuscript. My only comment is that you have not commented on the fastidiousness of N. Lactima in the laboratory i.e in terms of extraction, then growth then inoculation. What is the maximum turnaround time for this to occur before N. Lactima is no longer viable.
--

REVIEWER	Veeranoot Nissapatorn Walailak University, School of Allied Health Sciences
REVIEW RETURNED	11-Oct-2021

GENERAL COMMENTS	Dear authors, Thank you for submitting this manuscript. This clinical trial study is interesting esp. authors focus on horizontal transmission of mother-neonate pair. However, there are some points (all in yellow highlighted colors) to further improve this protocol to better understanding the overall concept of this study. Sincerely,
--

REVIEWER	Chun Sen Wu University of Southern Denmark, Research Unit of Gynecology and Obstetrics, Institute of Clinical Research
REVIEW RETURNED	14-Jan-2022

GENERAL COMMENTS	I think this is a well-written protocol and interesting topic. I have only a few comments. In the protocol, the authors mentioned “statistical analysis to compare mothers and infants” but no information on what statistical analyses will be used? in general, to avoid fishing, statistical analyses including any subgroup analysis and post-hoc analyses should be predefined in the protocol.
---

	The protocol described the aim (“The aim of this study is to establish if nasal inoculation with N. lactamica in pregnancy results in neonatal colonisation after birth, and to characterise the impact on the developing neonatal URT microbiome.”), the primary and secondary endpoints in the table 1. But it is unclear what authors’ hypotheses are? We may ask for a few potential questions, such as what is N. lactamica colonisation density between volunteers and their infants at one day after birth, one week, and one month postpartum, respectively? Or what is the expected N. lactamica colonisation density development overtime among volunteers and their infant? Based on the available knowledge what we expect/hypothesize to observe. Furthermore, in the table 1, regarding safety, it is unclear what “adverse reactions” and “serious adverse events” are. Without definition/specification, if I were a member of data safety monitoring board, it is difficult to evaluate the safety either during the study period or at the end of the study. By the way, have you consider establishing a data safety monitoring board? According to the protocol, “Enrolled volunteers who are already naturally colonised with N. lactamica will not be inoculated. However, they will be invited to remain in the study as an opportunistic comparison cohort, and will complete all post-partum visits and sampling as for the inoculated mother-neonate pairs”. Does the comparison cohort is additional or part of the sample size (N=20)? Through the protocol, it used “horizontal transmission” between mother and neonate. Should it be “vertical transmission” instead of “horizontal transmission”? In the table 1, please clarify what “sample types” mean? “Volunteers’ experiences of participation (including acceptability and tolerability of inoculation and maternal and infant sampling) will be captured using optional questionnaires”. The questionnaires should be attached to protocol. Very happy to see that “obtaining an umbilical cord rather than an infant venous blood sample at birth”. It is practically not easy to get venous blood sample among infants.
--	---

VERSION 1 – AUTHOR RESPONSE

Reviewer: 1

Dr. Melanie Bissessor, University of Melbourne, Melbourne Sexual Health Centre

Comments to the Author:

Thank you for the interesting manuscript.

My only comment is that you have not commented on the fastidiousness of *N. Lactima* in the laboratory i.e in terms of extraction, then growth then inoculation. What is the maximum turnaround time for this to occur before *N. Lactima* is no longer viable.

Many thanks for this helpful comment. We have added the following sentence to address this:

“The inoculum will be administered immediately following reconstitution and serial dilution, and after no more than 30 minutes’ delay; recovery of viable bacteria remains over 70% after 30 minutes incubation in room air (unpublished data).”

Reviewer: 2

Dr. Veeranoot Nissapatorn, Walailak University

Comments to the Author:

Dear authors,

Thank you for submitting this manuscript.

This clinical trial study is interesting esp. authors focus on horizontal transmission of mother-neonate pair. However, there are some points (all in yellow highlighted colors) to further improve this protocol to better understanding the overall concept of this study.

It would be much better if authors can provide an infographic figure of this overall study for better understanding. Otherwise it seems just black and white manuscript!

Thanks for that, we agree that an infographic is helpful here, and have added Figure 1.

NLAC inoculum Reference? If it was commercially purchased (company, city, country)

Many thanks for spotting that we hadn't included this! We've added the following clarification:

"The strain used for lyophilisation (Y92-1009, sequence type 3493, clonal complex 613) originates from the current GMP-compliant pharmaceutical manufacturing facilities at Public Health England (Porton Down, United Kingdom)."

mL instead of ml. Italics for genus/species

We have changed all of these as requested.

"Respiratory sampling to assess for baseline natural *N. lactamica* carriage": Please elaborate further, it is not clear to readers.

Hopefully the following explanation is clearer:

"A detailed medical history will be taken, followed by pharyngeal and saliva sampling to assess for *N. lactamica* carriage; enrolled volunteers who are already naturally colonised with *N. lactamica* will not be inoculated. However, they will be invited to remain in the study as an opportunistic comparison cohort, and will complete all post-partum visits and sampling as for the inoculated mother-neonate pairs (Table 2)."

List company, city, country:

We have added the details requested for API-NH (bioMérieux, Lyon, France), GC agar (Thermo Fisher Scientific, Basingstoke, United Kingdom) and MALDI-TOF (Bruker, Coventry, United Kingdom).

Please provide more details on what type of PCR use (conventional, RT etc) with a brief protocol.

Many thanks, we have added a paragraph describing our PCR, and have also included a reference to a detailed book chapter our team has recently published, describing the primers and methods in more detail:

"We have identified two chromosomal loci present in *N. lactamica* strain Y92-1009 that, when amplified in conventional multiplex PCR and analysed by agarose gel electrophoresis, produce amplicons with a characteristic band pattern¹⁸. Using the same multiplex PCR on isolates of naturally circulating strains of *N. lactamica*, this band pattern has not been reproduced. A third locus, present in all known strains of *N. lactamica*, is included to differentiate between PCR failure (no bands produced) and any non-Y92-1009 strain of *N. lactamica* (band produced, but Y92-1009-specific bands pattern not present). This PCR assay has been validated using genomic DNA and boiled bacterial lysate as a template, with an optimum annealing temperature 65°C and extension time 15 seconds (unpublished data)."

Reviewer: 3

Dr. Chun Sen Wu, University of Southern Denmark, Odense University Hospital

Comments to the Author:

I think this is a well-written protocol and interesting topic. I have only a few comments.

In the protocol, the authors mentioned “statistical analysis to compare mothers and infants” but no information on what statistical analyses will be used? in general, to avoid fishing, statistical analyses including any subgroup analysis and post-hoc analyses should be predefined in the protocol.

Thank you for your consideration and helpful comments. We have added the following statement on statistical analysis:

“The primary endpoint will be expressed as a proportion N/M, where:

- M = Number of mothers colonised with *N. lactamica*;
- N = Number of neonates (aged 0-31 days) colonised with the same strain of *N. lactamica* as their own mother;
- “Colonised” = evidence of *N. lactamica* by culture of upper respiratory swabs (using selective media and confirmation by API or MALDI-TOF) on at least one visit;
- “Same strain” = confirmed using Y92-specific *N. lactamica* PCR for inoculated volunteers, or whole genome sequencing for volunteers already naturally carrying *N. lactamica* at screening.

Subgroup analysis will be performed, to compare inoculated mother-infant pairs with naturally colonised mother-infant pairs, and to compare pairs where the mother or infant received antibiotics with pairs not receiving antibiotics.

Specific bioinformatic analysis strategies will depend on the quantity and quality of genomic data produced during the study, and on the rates of *N. lactamica* colonisation observed. Broadly speaking, microbiome bioinformatic pipelines (e.g. QIIME2 and R packages) will be used to perform taxonomic classification, calculation of alpha and beta diversity, and data visualisation.”

*The protocol described the aim (“The aim of this study is to establish if nasal inoculation with *N. lactamica* in pregnancy results in neonatal colonisation after birth, and to characterise the impact on the developing neonatal URT microbiome.”), the primary and secondary endpoints in the table 1. But it is unclear what authors’ hypotheses are? We may ask for a few potential questions, such as what is *N. lactamica* colonisation density between volunteers and their infants at one day after birth, one week, and one month postpartum, respectively? Or what is the expected *N. lactamica* colonisation density development overtime among volunteers and their infant? Based on the available knowledge what we expect/hypothesize to observe.*

Many thanks for that feedback, we have now included a hypothesis statement before describing our objectives and endpoints:

“We hypothesise that nasal inoculation of pregnant women with 10⁵ CFU *N. lactamica* strain Y92-1009 will result in maternal upper respiratory *N. lactamica* colonisation; and that horizontal transmission of upper respiratory *N. lactamica* will occur from experimentally-colonised women to their infants by 4 weeks post-partum.”

Furthermore, in the table 1, regarding safety, it is unclear what “adverse reactions” and “serious adverse events” are. Without definition/specification, if I were a member of data safety monitoring board, it is difficult to evaluate the safety either during the study period or at the end of the study.

Thank you, we’ve now clarified this:

“As per the current Good Clinical Practice guideline²², adverse events are any untoward medical occurrence in a volunteer or their neonate, temporally, but not necessarily causally, related to study participation. Delivery of routine peripartum care is not considered an adverse event. Serious adverse events include hospitalisation (or prolongation of hospitalisation for routine peripartum care), persistent or significant disability, a life-threatening occurrence, or death from any cause.”

By the way, have you consider establishing a data safety monitoring board?

Yes, we do have an External Safety Committee, which we refer to in the Ethics and Safety section of the manuscript.

According to the protocol, “Enrolled volunteers who are already naturally colonised with *N. lactamica* will not be inoculated. However, they will be invited to remain in the study as an opportunistic comparison cohort, and will complete all post-partum visits and sampling as for the inoculated mother-neonate pairs”. Does the comparison cohort is additional or part of the sample size (N=20)?

Apologies if this wasn’t clear. The target of 20 inoculations does not include natural carriers (as these participants will not be inoculated), hence the anticipated number of enrolled participants is higher than the target inoculation group. This is explained in the Sample Size subsection.

Through the protocol, it used “horizontal transmission” between mother and neonate. Should it be “vertical transmission” instead of “horizontal transmission”?

Thank you for bringing up this interesting point. We do not anticipate transmission of *N. lactamica* from mother to infant to occur transplacentally in pregnancy or transvaginally at the time of delivery, but rather by direct upper respiratory mucosal contact after birth. So we therefore think that horizontal transmission is a more appropriate term here than vertical transmission.

In the table 1, please clarify what “sample types” mean?

Thanks for pointing that out. We have changed to “samples types” to “sample niche” for clarity (i.e. saliva, nasopharyngeal, or oropharyngeal).

“Volunteers’ experiences of participation (including acceptability and tolerability of inoculation and maternal and infant sampling) will be captured using optional questionnaires”. The questionnaires should be attached to protocol.

Please find both questionnaires attached now.

VERSION 2 – REVIEW

REVIEWER	Veeranoot Nissapatorn Walailak University, School of Allied Health Sciences
REVIEW RETURNED	25-Mar-2022

GENERAL COMMENTS	The proposal is supported for publication.
--

REVIEWER	Chun Sen Wu University of Southern Denmark, Research Unit of Gynecology and Obstetrics, Institute of Clinical Research
REVIEW RETURNED	21-Apr-2022

GENERAL COMMENTS	Nice written protocol! In the line 189 mentioned “Safety outcomes”, in the line 205-206 mentioned “safety profile”, and in the table 1, mentioned “Safety endpoints”, however, it is still unclear what the terms exactly are. Please clarify and define the terms. As described in the line 205-206 “This study aims to investigate this hypothesis, and to characterize the safety profile,” As the aim of the project is to characterize the safety profile, which must be clearly defined, right?
---